# A New Plasticity Index Including Size-Effects in the Contact of Rough Surfaces

## M. Ciavarella

Center of Excellence in Computational Mechanics, Politecnico di BARI, Viale Japigia 182, 70126 Bari, Italy; mciava@poliba.it

**Abstract:** It is known that contact of rough surfaces occurs over an area much smaller than the nominal contact area, and at asperity scale, increased hardness results in experimentally observed asperity "persistence", namely that it is hard to flatten asperities. Here, we consider Persson's elasto-plastic solution for rough contact together with an hardness equation proposed by Swadener, George and Pharr for spherical indentation, including size effects depending on sphere radius, in particular to define a new plasticity index that defines the tendency to plastic deformation. While the classical plasticity index shows that at sufficiently small scales, there will be plastic deformations unless surfaces are extremely smooth, and with size effects, the small roughness scales the content of spectrum matter in defining the real state of asperities. In particular, what may appear as plastic at a bulk scale returns to an elastic behaviour at a small scale, as suggested by the "asperity persistence" experimental observation. Some illustrative examples are shown, but clearly, our index and elasto-plastic solution are mainly qualitative, as a realistic investigation is much more complex and still computationally too demanding.

**Keywords:** rough contact; Persson's solution; hardness; indentation size effects





## 1. Introduction

It has been known for a long time that in the contact of metallic bodies, due to roughness, the real contact area is much smaller than the nominal contact one, unless the load is exceptionally high. However, even when the load is very high, like in a hardness test, so the bulk of the material does deform plastically and the nominal contact area increases (nearly) linearly with the load, even in a spherical geometry, the asperities do not flatten out perfectly. Only some fraction at the top of the asperities actually flattens, and classical experiments suggested that this part was a certain percentage of the projected macroscopic indentation area [1–4]. One possible explanation for the phenomenon, called the "persistence of asperities", is the increase in indentation hardness at smaller length scales, which contradicts the classical theory of plasticity, but which has been reported in micro- and nanoindentation experiments and explained by several authors using concepts of density of dislocations or strain gradient plasticity [5–7]. Indeed, hardness measured in crystalline materials with sharp indenters increases with decreasing depth of penetration, whereas for spherical indentation, the relevant length scale is the radius of asperity. In spherical indentation, then, it has been suggested by Swadener, George and Pharr that hardness, which at macroscopic scale appears as $H_0$, increases with decreasing sphere radius $R$ as $H_y(R)$

$$\frac{H_y(R)}{H_0} = \sqrt{1 + \frac{R^*}{R_p(R)}} \tag{1}$$

where $R_p$ is the radius of the impression, which is 10–20% larger than the radius of the asperity $R$, so we can assume $R_p = 1.2R$ for the purpose of our discussion, and $R^*$ is written strictly in terms of material constants [7]. Some authors suggest that at nanoscales, the

hardness would be overpredicted, and alternative theories exist, for example, with a maximum allowable density of geometrically necessary dislocations [8]—but for its simplicity, the Swadener, George and Pharr law is the most simple assumption for our purposes.

It should be said that other mechanisms may be at play to justify the increase in hardness, one being interaction between asperities, studied, for example, by Gao and Bower [9], who found that this could double the typical hardness Tabor prediction of $H_0 = 3\sigma_y$ (where $\sigma_y$ is the yield stress in uniaxial conditions), and Manners [10] showed that there would be no finite limit to this increase in a rigid perfectly plastic model in a situation of full containment of plastic flow. Yet, another mechanism is work hardening, which is, however, relevant only for some materials, and generally is governed, for a spherical indentation, by the strain level expressed as $a/R$, where $a$ is contact radius [7]. Generally size effects tend to be more pronounced than hardening effects, and we shall consider mainly the former.

In recent years, significant attention has been paid to the description of contact bodies with rough surfaces. Persson introduced a successful theory of elastic contact between randomly rough surfaces [11], where the probability distribution of contact pressures is found to evolve as more wavelengths of roughness are introduced, resulting in a linear diffusion equation for the probability distribution. The theory was mostly developed for elastomers, for which very high strains can be reached in the elastic regime (although nonlinearity may emerge). A much less known generalization of the theory to the case of elasto-plastic materials exists [12], which could be more appropriate for metallic materials, and is perhaps simplistic as it merely replaces a boundary condition on his linear diffusion equation so that no pressure can be higher than the hardness limit. Persson also attempted to introduce the size-dependence of hardness as a function of the "magnification" (the ratio of the upper wavevector cutoff $q_1$ to lower wavevector cutoff $q_0$ in the roughness power spectrum PSD) without specific reference to actual possible dependencies [13], while Xu et al. [14] obtained a closed-form result for the elasto-plastic theory (which we summarize in Appendix A, as we use here). The theory can only give the area–load relationship, and not the load–displacement one. Venugopalan et al. [15] attempted to compare Persson's elasto-plastic theory with full dislocation dynamics simulations, and found the results to be in good agreement when the rough surface has a very small root-mean-square (rms) height. This is rather expected, since in that case, the contact is mainly elastic, and Persson's theory is known to work well in this case. For larger and *more realistic* rms heights for metal surfaces, the agreement is no longer good if one uses a size-independent hardness, so Venugopalan et al. [15] introduced the yield strength found directly in the simulation, which makes the agreement good again, but quite obvious. Venugopalan et al. [15] obtain that the pressure on the asperities is low for the low $h_{rms}$ case (and not far from the reference hardness without size effect), while it is high (*10 time higher*) for the large $h_{rms}$ where the radius of asperities will be smaller by way of how the construct roughness with same upper and lower wavevector cutoffs, in qualitative agreement to our view of the Swadener, George and Pharr law (our Equation (1)).

Tiwari et al. [16] conduct an interesting experimental campaign of indentation experiments of aluminum blocks, of which they measure roughness before and after spherical indentation, and compare them with a BEM numerical simulations where elastic deformations are exact but plastic deformations are simply obtained without restricting perfectly plastic flow without work hardening. As size-independent indentation hardness leads to a spherical-cup indented area with all asperities flattened, contradicting the experiments, Tiwari et al. [16] increase indentation hardness at the asperity level by a factor of about 2.5—ad hoc for their case. The deformation predicted by the simple BEM model then agrees well (at least, qualitatively) both in terms of the strongly skewed height probability distribution and the surface roughness power spectra of the plastically deformed surfaces. However, results that seem valid for their aluminum may not be general, as indeed they recall that for some polymers, in particular polyethylene, after plastic deformation,

a perfectly symmetric Gaussian-like height probability distribution was experimentally previously found.

Jackson and Jacobs [17] recently discussed multiscale and statistical models, showing that the inclusion of scale-dependent strength leads to a prediction of contact area closer to that of the elastic model, particularly when a wide range of size scales is included. Violano and Afferrante [18] use a model of interacting and coalescing spherical asperities, so that they can use well-known previous results of indentation of elasto-plastic spheres, but neglect size effects. They find, as expected, that the nanoscale amplitude of roughness can correspond to the elastic solution, but for microscopic scale roughness, plasticity is important, and a good approximation for the area–load relationship is Persson's elasto-plastic solution, which converges when adding smaller wavelengths in the roughness due to the constant hardness.

In conclusion of this brief overview, we see that the existing literature strongly differs in the conclusion, depending on the assumption of size effects of the hardness, with simpler models without size effects leading to plasticity at small scales, while more refined models, which include size effects, show a strong increase in asperity pressures, and therefore hardness, and a behavior close to the elastic one.

Returning to more classical work, one indicator for the emergence of plasticity in the contact of rough surface of rms roughness $h_{rms}$ was introduced by Greenwood and Williamson [19] whose model was based on identical (and independent) asperities of radius $R$, leading to an index

$$\Psi_{GW} = \frac{E}{H_0}\sqrt{\frac{h_{rms}}{R}} \tag{2}$$

where $E$ is the (composite) elastic modulus of the contacting surfaces. Large roughness or small asperity radius contribute to a higher likelihood of plastic deformations, indicated by $\Psi_{GW} > 1$.

In the present paper, we shall combine Persson's elasto-plastic solution with the Swadener, George and Pharr law (Equation (1)) and introduce an appropriate plasticity index as a consequence. The state of asperities in the contact of (metallic) rough surfaces is a problem which is still largely unresolved but is relevant to a wide spectrum of applications in tribology, starting from friction. While the classical Bowden and Tabor theory of friction is based on a plastic state of asperities, already classical experiments and theory by Archard [20] suggested that Amonton's law for friction could hold also with an *elastic* multiasperity model especially for loading after a phase of running in. Lim et al. [21] do report for poorly lubricated metallic contact at low speeds a friction coefficient which increases with roughness. Tabor [22–24] discusses the implications of the classical plasticity index for the basic understanding of friction wear and adhesion. In particular, he argues that for most engineering surfaces, the classical plasticity index is greater than 1, the true contact pressure is the hardness, and the contact is predominantly plastic. In particular, Tabor recognizes that only for bearing steel (which is quite hard), the contact is predominantly elastic even for relatively rough surface. Even Tabor does discuss the effect of size in the yield strength of material, mentioning for example that gold had been observed to have 10 times higher strength at small scales, although this increase may be reduced for work hardened materials. However, at the time of Tabor there was no theoretical model to take this size effect into consideration. Unfortunately, there are many difficulties to make the connection between the plasticity indexes and actual tribological performances quantitative: the plasticity index is not related quantitatively to any macroscopic quantity such as wear or friction coefficient, and hence it is difficult to make quantitative judgements on what should be its correct form. Only recently numerical and experimental techniques are starting to give some progress on the real conditions at asperity levels (and we have given some examples), and would lead in the future to a better understanding at this scale, and make connection with the macroscopic scale in quantitative terms.

Our new index and our discussion is mainly aimed at prompting more discussion on rough contacts and guiding full-scale simulations, which are emerging recently; see, for

example, the case of repeated indentation [25], which in the future will be able to include broad spectra of roughness, size effects, and other realistic features in the plastic behavior. A plasticity index is certainly a concise measure that we hope to have as an alternative to full-scale simulations or very sophisticated experimental investigations, and this gives the motivation of the present paper. Notice that we are looking at the size effect in hardness, while we are assuming dry contact, isothermal conditions, and an absence of adhesive effects, and we are neglecting strain hardening.

## 2. The Model

Assume the surface $h(x, y)$ has a continuous noise spectrum in two dimensions and is described by a Gaussian stationary process. In such case, we write

$$h(x, y) = \sum_n C_n \cos\left[q_{x,n} x + q_{y,n} y + \phi_n\right] \tag{3}$$

where the wave-components $q_{x,n}$ and $q_{y,n}$ are supposed to be densely distributed throughout the $(q_x, q_y)$ plane. The random phases $\phi_n$ are uniformly distributed in the interval $(0.2\pi)$. The amplitudes $C_n$ are also random variables. The function $C(q_x, q_y)$ is the power spectral density (PSD) of the surface $h$. Defining in general for the $(s, t)th$ moment of $C(q_x, q_y)$ as

$$m_{st} = \int_{-\infty}^{+\infty} \int_{-\infty}^{+\infty} C(q_x, q_y) q_x^s q_y^t dq_x dq_y \tag{4}$$

In particular, one can show that the rms roughness $h_{rms} = \left\langle h^2 \right\rangle = m_0$, while the root-mean-square slope of the surface is

$$h'_{rms} = \sqrt{\left\langle |\nabla h|^2 \right\rangle} = \sqrt{\left\langle \left(\frac{\partial h}{\partial x}\right)^2 + \left(\frac{\partial h}{\partial y}\right)^2 \right\rangle} = \sqrt{m_{20} + m_{02}} = \sqrt{2m_2} \tag{5}$$

Finally, if we define the rms curvature as $h''_{rms} = \sqrt{\left\langle (\nabla^2 h)^2 \right\rangle}$, then

$$h''_{rms} = \sqrt{\left\langle \left(\frac{\partial^2 h}{\partial x^2} + \frac{\partial^2 h}{\partial y^2}\right)^2 \right\rangle} = \sqrt{\left\langle \left(\frac{\partial^2 h}{\partial x^2}\right)^2 + \left(\frac{\partial^2 h}{\partial y^2}\right)^2 + 2\frac{\partial^2 h}{\partial x^2}\frac{\partial^2 h}{\partial y^2} \right\rangle}$$
$$= \sqrt{m_{40} + m_{04} + 2m_{22}} = \sqrt{8m_4/3} \tag{6}$$

Let us consider, for simplicity, a roughness for a self-affine surface of pure power law isotropic PSD (power spectrum density) $C(q) = Zq^{-2(1+H)}$ for $q_1 > q > q_0$, $[C] = [m^4]$, $[Z] = [m^{2-2H}]$ where $H$ is the Hurst exponent, not to be confused with hardness $H_y$. The moments of the spectrum are therefore

$$m_0 \simeq \frac{\pi}{H} Z q_0^{-2H}, \qquad m_2 \simeq Z\pi q_0^{2-2H}\left(\frac{\zeta^{2-2H}}{2-2H}\right) \tag{7}$$

$$m_4 \simeq Z\frac{3}{4}\pi q_0^{4-2H}\left(\frac{\zeta^{4-2H}}{4-2H}\right) \tag{8}$$

The mean radius of asperities (summits) from random process theory is therefore

$$R_s = \frac{3}{8}\sqrt{\frac{\pi}{m_4}} \tag{9}$$

The mean pressure at the asperity level at a given spectrum breadth (or "magnification" in Persson's notation), $\zeta = q_1/q_0$ where $q_1$ is the largest wavevector introduced in the

spectrum and $q_0$ is the smallest, in the elastic Persson model for not too large areas ($A/A_0 < 0.3$–0.4), is

$$\overline{p}_{asp} = \frac{P}{A} = \sqrt{V} = \frac{1}{2} E h'_{rms} = \frac{1}{2} E \sqrt{\frac{\pi Z}{1-H}} q_1^{1-H} \tag{10}$$

where $A$ is the real (elastic) contact area (not the nominal one $A_0$) and $\sqrt{V} = \frac{1}{2} E h'_{rms}$ is the rms full contact pressure. In the present paper, we do not provide a criterion for when a single asperity changes its behavior from elastic to plastic. For any value of plasticity index, the mean pressure at the asperity level is constant and load-independent: although some asperities will have increased load if macroscopic load is increased, new asperities are formed and the average remains the same. Therefore, in this sense, following the exact behavior of each asperity is not needed. Hence, the ratio of mean asperity pressure to hardness defines a "plasticity index":

$$\Psi_P = \frac{\overline{p}_{asp}}{H_0} = \frac{\sqrt{V}}{H_0} = \frac{E}{2H_0} \sqrt{2m_2} = \frac{1}{2} \frac{E}{H_0} \sqrt{\frac{\pi Z}{1-H}} q_1^{1-H} \tag{11}$$

and hence since for metals $\frac{E}{H_0} = 25 - 1000$ [26], it seems that we need extremely smooth roughness to avoid plasticity (since at small scales it is not uncommon to find $\sqrt{m_2} \simeq 1$), neglecting indentation size effects. For example, the order of roughness reported in [18] to be in the range between the elastic and plastic regime with magnification of $\zeta = 128$ is in the order of 30 nanometers, which is unlikely to be the surface finish of metallic rough surfaces even after the process of running in. Hence, the standard case with macroscopic hardness would show some tendency to plastic deformation at the asperity scale, and it becomes important, therefore, to include indentation size effects if we want to speculate more accurately.

A new plasticity index can therefore be formulated considering indentation size effects by combining the *elastic* Persson's theory with the Swadener, George & Pharr law (Equation (1)) obtaining

$$\Psi^* = \frac{E}{2H_0 \sqrt{1 + \frac{R^*}{1.2\frac{3}{8}\sqrt{\frac{\pi}{m_4}}}}} \sqrt{2m_2} \tag{12}$$

Assuming asperity radius is small enough so that $\frac{R^*}{R_p} >> 1$ and the case of pure power law spectrum, we obtain

$$\Psi^* \simeq 0.335 \frac{E}{H_0 \sqrt{R^*}} \frac{2^{1/2} \pi^{1/2} (4-2H)^{1/4}}{\left(\frac{3}{4}\right)^{1/4} (2-2H)^{1/2}} Z^{1/4} q_1^{-H/2} \tag{13}$$

and hence the new index (13) consists of two terms: a material parameter $\frac{E}{H_0 \sqrt{R^*}}$ and a geometrical roughness-related quantity, which is low for low power spectrum intercept $Z$ and high large wavevector cutoff $q_1$. Both the original index $\Psi_P$ (Equation (11)) and the new index $\Psi^*$ (Equation (13)) seem to depend crucially on how we truncate the roughness spectrum. However, while $\Psi_P$ (Equation (11)) grows as $q_1^{1-H}$, the new index $\Psi^*$ (Equation (13)) *decays* as $q_1^{-H/2}$; therefore, while we can obtain that for sufficiently high "magnification", the old index indicates that all asperities smaller that a certain scale are in a plastic regime, the next index could indicate that contact may *appear* as plastic at a bulk scale of large asperities, but *returns to an elastic behavior at small scales*, in agreement with the "asperity persistence" experimental observation.

## 3. Some Results Using the Persson Elasto-Plastic Solution

The plasticity index we have defined is based on the elastic classical solution from Persson. However, the elasto-plastic solution suggests that there is always a certain per-

centage of the nominal contact area that is under plastic contact. Simply, this percentage becomes significant when the rms full contact pressure is in the same order of hardness, and therefore when the plasticity index is 1. Any use of the elasto-plastic solution to define an alternative plasticity index would be based anyway on a choice of a threshold of plastic contact area, and hence would not be more precise.

Let us start with the case of the absence of indentation size effects. Since the solution for low loads is linear, in the plastic regime, we can also write in this case simply

$$\frac{A}{A_0} = \frac{\overline{p}}{H_0} \tag{14}$$

whereas the original elastic Persson's solution is

$$\frac{A}{A_0} = \frac{A_{el}}{A_0} = \sqrt{\frac{2}{\pi}} \frac{\overline{p}}{\sqrt{V}} \tag{15}$$

Expressions for the total contact area in the Persson plastic solution are given in Appendix A, and the plastic contact area can therefore be obtained by subtracting the elastic one. As we discussed in the introduction, even very smooth surfaces would show significant plastic deformations for metals. We consider, for example, a very smooth surface having $q_0 = 10^5 \times m^{-1}$ , $H = 0.8$ and $Z = 10^{-8} \times m^{0.4}$ resulting in $h_{rms} = \sqrt{m_0} \simeq \sqrt{\frac{\pi}{H}Zq_0^{-H}} = \sqrt{\frac{\pi}{0.8}10^{-5}}10^{-5\times0.8} = 20$ nm while $\zeta = q_1/q_0 = 128$. For this case, we show the results of Figure 1, showing that the total contact area (red solid line) as the ratio of hardness to elastic modulus $H_0/E$ increases, transitions from coinciding with the fully plastic one (the blue solid line) to the fully elastic Persson's elastic one (green line). However, only because the surface is very smooth and the "magnification" is quite low (smallest wavelengths on the micron scale $\lambda_1 = 2\pi/q_1 = 2\pi/\left(128 \times 10^5\right) = 0.5$ μm) do we find that for practically the entire realistic range of hardness, we observe an elastic solution.

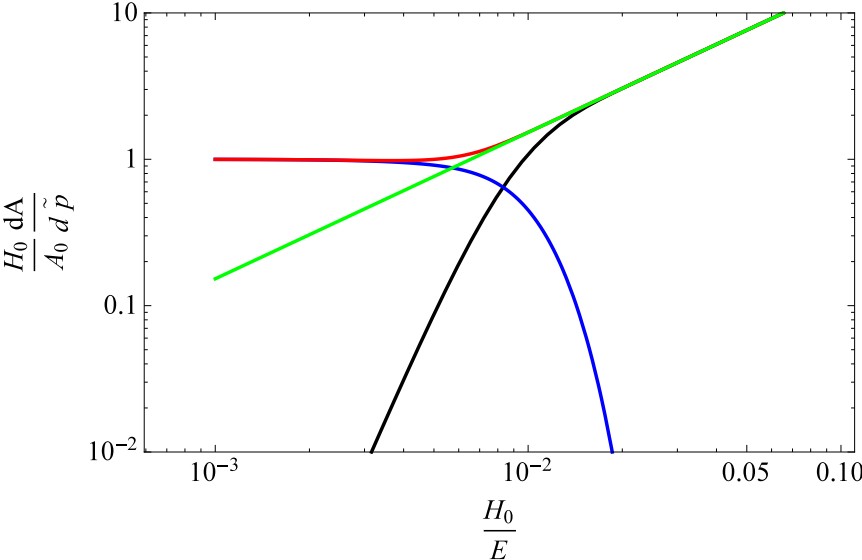

**Figure 1.** The normalized initial slope of the area–load relationship $\frac{H_0}{A_0}\frac{dA}{d\overline{p}}$ as a function of hardness to elastic modulus $H_0/E$ for a rough surface, indicated in the text with $h_{rms} = 20$ nm and with magnification of $\zeta = 128$. Solid black line indicates elastic area, blue line the plastic area, red line the total contact area. Green line is the elastic Persson solution.

However, if we now fix the hardness to a quite high value $H_0/E = 10^{-2}$ and expand the magnification $\zeta = q_1/q_0$ to various values to see the effect of a wider and wider spectrum of scales, we obtain Figure 2. Clearly, we see that even with such a smooth surface

(in the sense of low $h_{rms}$, which is hardly affected by $\zeta = q_1/q_0$), the inclusion of large wavevectors (small scales content) implies a transition towards the plastic regime if we consider asperities in the order of $\lambda_1 = 2\pi/(1000 \times 10^5) = 60$ nm. Here, such a model would imply no asperity persistence.

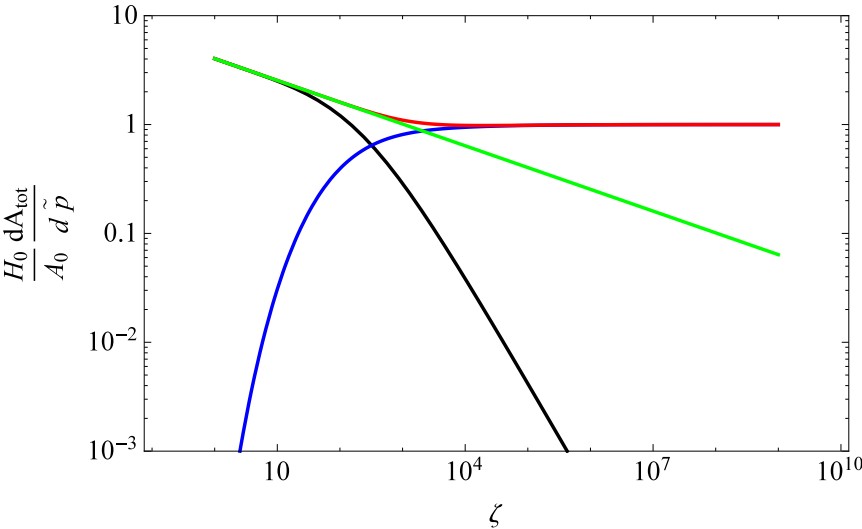

**Figure 2.** The normalized initial slope of the area–load relationship $\frac{H_0}{A_0}\frac{dA}{d\widetilde{p}}$ as a function of magnification $\zeta = q_1/q_0$ for a rough surface indicated in the text with $h_{rms} = 20$ nm and material with high hardness to a quite high value $H_0/E = 10^{-2}$. Solid black line indicates elastic area, blue line the plastic area, red line the total contact area. Green line is the elastic Persson solution.

### 3.1. Persson's Solution Modified with Size Effects

If we now consider the effect of size-dependent hardness, we obtain for the same case of Figures 1 and 2 with macroscopic hardness $H_0/E = 10^{-2}$ and with $R^* = 1$ mm, the results of Figure 3 where the area–load slope is now given by the elastic solution in the entire range of magnifications. The lines of total contact area overlap with those of the elastic Persson's solution, while the plastic area is zero in the entire range.

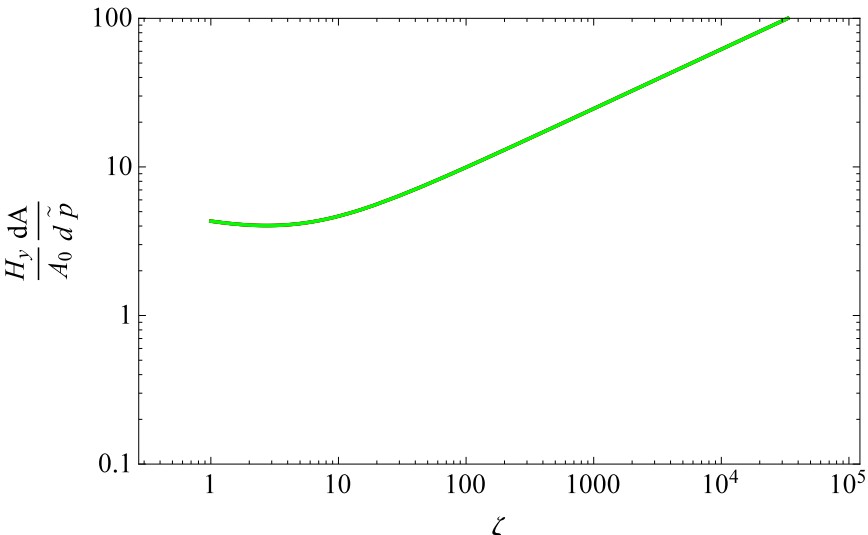

**Figure 3.** Same as Figure 2, but here, normalization is with actual size-dependent hardness $\frac{H_y}{A_0}\frac{dA}{d\widetilde{p}}$, so that it first decreases and then increases with magnification $\zeta$.

In fact, thanks to size-dependent hardness, we can increase largely the rms amplitude of roughness to more realistic ranges, and at the same time take an example with decreased

macroscopic hardness, yet find asperity-level elastic behavior. For example, we take $H_0/E = 10^{-3}$ and $h_{rms} = 600$ nm, with all parameters remaining the same, obtaining the results of Figure 4, which return to elastic behavior at magnification of about $\zeta = 10^4$.

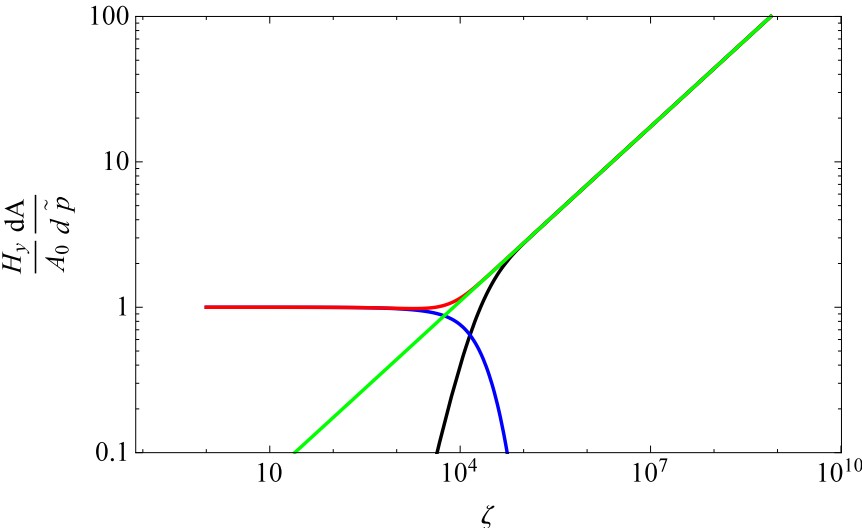

**Figure 4.** Same as Figure 3, but here with increased amplitude of roughness and decreased hardness, namely $H_0/E = 10^{-3}$ and $h_{rms} = 600$ nm.

### 3.2. Discussion

Independently of size effect, due to hardening, hardness is also dependent on the ratio $\varepsilon = a/R$. Individual contacts are not circular in real rough contacts. With the Pastewka and Robbins [27] full numerical model, which finds a repulsive diameter, which is independent on load—a characteristic size that they estimate depends only on purely geometrical quantities:

$$d_{rep} = 4h'_{rms}/h''_{rms} \qquad (16)$$

The total contact area in the elastic regime then is obtained by multiplying this diameter by a perimeter, resulting in a contact area that is fractal in character both for area and for perimeter. Hence, the ratio $\varepsilon = a/R$

$$\varepsilon = \frac{d_{rep}}{R} = 4\frac{h'_{rms}}{h''_{rms}}\frac{h''_{rms}}{2} = 2h'_{rms} = \sqrt{\frac{2Z\pi}{2-2H}}q_1^{1-H} \qquad (17)$$

increases with wavevector cutoff, like indentation size effects, but it is unclear how work-hardening effects and size effects combine, namely if they are sinergistic or the combined effect is smaller than the cumulative effect of each of the two in isolation.

It is clear that the exact state of asperities on one hand is difficult to assess either by numerical or by experimental methods, and on the other may not be that relevant when we consider the macroscopic result on the laws of tribology. Indeed, as we have recalled in the introduction, classical models for friction explain Amonton's law (linear dependence of frictional force on normal load) both with a plastic state of asperities (Bowden and Tabor's theory) or with an elastic one (Archard). The quantitative value of the friction coefficient for some metals and dry friction conditions is also close to what was suggested by Bowden and Tabor, with the simple view that it should correspond to the ratio of the shear strength to the yield strength, which substantiates the prevailing view of the plastic theory. However, a full plastic model also seems to suggest the highest damage to surfaces and the highest wear, whereas an *elastic* multiasperity model may be closer to the real state of the contact, especially for loading after a phase of running in. Lim et al. [21,28] report for poorly lubricated metallic contact at low speeds a friction coefficient that increases with roughness.

Obviously, our new index and our discussion are mainly aimed at prompting more discussion on rough contacts and guiding full-scale simulations, which are emerging recently; see, for example, the case of repeated indentation, which in the future will be able to include broad spectra of roughness, size effects and other realistic features in plastic behavior. A plasticity index is certainly a concise measure that we hope to have as an alternative to full-scale simulations or very sophisticated experimental investigations, and this gives the motivation of the present paper. Notice that we are looking at the size effect in hardness, while we are assuming dry contact, isothermal conditions, and an absence of adhesive effects, and we are neglecting strain hardening.

## 4. Conclusions

We have discussed the role of indentation size effects in the contact of rough surfaces, combining the Swadener, George and Pharr law (Equation (1)) into Persson's elastic solution to define a new plasticity index $\Psi^*$ (Equation (13)), which depends on material properties, including a characteristic radius of asperities, below which the size effect becomes important, and geometry dependent quantities. Contrary to classical plasticity index, the new index shows that small-scale features may return to elastic behavior, even though at a macroscale, contact shows some bulk plasticity.

Therefore, it becomes important to study the plasticity at different scales. Obviously, our index is merely a combination of simple equations, and may serve as a rapid assessment of rough contacts, which may be useful, because the full investigation, involving a broad spectra of roughness with discrete dislocation models or strain gradient plasticity, seems to date to be computationally very intensive, although some examples exist [15,29] that are in line with the results of the present investigation.

**Funding:** This research received no external funding.

**Data Availability Statement:** Data are available upon reasonable request from the author.

**Conflicts of Interest:** The authors declare no conflicts of interest.

## Nomenclature

| | |
|---|---|
| $h(x,y)$ | surface elevation of the rough surface |
| $C(q_x, q_y)$ | the power spectral density (PSD) of the surface $h$ b$R$ radius of asperity |
| $H_y(R)$ | hardness dependent on radius of asperity |
| $H_0$ | macroscopic-level hardness |
| $R^*$ | characteristic length scale |
| $R_p$ | the radius of the indentation impression, 10–20% larger than the radius of the asperity $R$ |
| $\sigma_y$ | the yield stress in uniaxial conditions |
| $a$ | contact radius |
| $q_1$ | upper wavevector cutoff in the roughness power spectrum PSD |
| $q_0$ | lower wavevector cutoff in the roughness power spectrum PSD |
| $h_{rms}$ | rms surface roughness height |
| $h'_{rms}$ | root-mean-square slope of the surface |
| $\Psi_{GW}$ | Greenwood–Williamson plasticity index |
| $E$ | the (composite) elastic modulus of the contacting surfaces |
| $h''_{rms} = \sqrt{\left\langle \left(\nabla^2 h\right)^2 \right\rangle}$ | rms curvature of the surface |
| $H$ | Hurst exponent of the surface |
| $m_0, m_2, m_4,$ | moments of the surface spectrum |
| $Z$ | amplitude of the surface power spectrum |

## Appendix A. Persson's Elasto-Plastic Solution

We summarize here the Persson elasto-plastic solution in the form suggested by Xu et al. [14]. We start by recalling the variance of full contact pressures

$$V = \frac{E^2}{2} m_2 \tag{A1}$$

where $E$ is plane strain composite modulus of the material pair, and $m_2$ is the second moment of the roughness spectrum.

We define the basic Gaussian

$$P_0(p, V) = \frac{1}{\sqrt{2\pi V}} \exp\left(-\frac{p^2}{2V}\right) \tag{A2}$$

and the two coefficients

$$a = \frac{P_0(H - \overline{p}, V)[P_0(\overline{p}, V) - P_0(2H - \overline{p}, V)]}{P_0(H - \overline{p}, V)P_0(\overline{p}, V) - P_0(H + \overline{p}, V)P_0(2H - \overline{p}, V)}$$

$$b = \frac{P_0(\overline{p}, V)[P_0(H - \overline{p}, V) - P_0(H + \overline{p}, V)]}{P_0(H - \overline{p}, V)P_0(\overline{p}, V) - P_0(H + \overline{p}, V)P_0(2H - \overline{p}, V)}$$

Finally, the total area $A_{tot}(V)$ is the sum of the elastic and plastic area $A_{el}$, $A_{pl}$, and with respect to the nominal contact area $A_0$, gives

$$\frac{A_{tot}(\overline{p}, V, H)}{A_0} = \frac{1 + a}{2} \operatorname{erf}\left(\frac{\overline{p}}{\sqrt{2V}}\right) + b \operatorname{erf}\left(\frac{\overline{p} - 2H}{\sqrt{2V}}\right) + \frac{1}{2}(1 - a + b) \tag{A3}$$

The classical Persson elastic solution mentioned in the introduction paragraph

$$\frac{A_{el}}{A_0} = \frac{\overline{p}}{\sqrt{V}} = \frac{2\overline{p}}{E\sqrt{2m_2}} \tag{A4}$$

is slightly different from that obtained from Persson's full solution

$$\lim_{\overline{p} \to 0} \frac{A_{el}}{A_0} = \lim_{\overline{p} \to 0} \operatorname{erf} \frac{\overline{p}}{\sqrt{2V}} \to \frac{\sqrt{8}}{\sqrt{\pi}} \frac{\overline{p}}{E\sqrt{2m_2}} \tag{A5}$$

as is a small improvement considering numerical solutions.

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
