# Peer review of "A New Plasticity Index including Size-Effects in the Contact of Rough Surfaces"

_lubricants, doi:10.3390/lubricants12030083_

Round 1
Reviewer 1 Report
Comments and Suggestions for Authors
This is a nice and concise paper. I enjoyed reading through it.
I have a few suggestions which I hope would of use to the author to improve the paper even further.
This is an opportunity to really make a definitive contribution regarding plasticity index and its use in description of nature of rough surfaces and their propensity to wear. So:
1- Slightly more detailed historical review would be nice. The current review is quite nice, but perhaps some of the works/opinions of Tabor would enhance it. For example please see:
Tabor D., “A simplified account of surface topography and the contact between solids”, Wear, 1975, 32(2):269-271.
Tabor, D., "Friction—the present state of our understanding", J. of Lubrication Tech. Apr 1981, 103(2), pp. 169-179.
In particular the role of adhesion in frictional behaviour and how plasticity index can be a useful measure:
Tabor, D., “Interaction between surfaces: adhesion and friction”, Surface Physics of Materials, 1974,11:475-529.
This last paper was very timely as it came along hot on the heal of JKR and DMT models, which can get a mention as well if adhesion issue is also discussed to some extent.
Point to remember is that with increasingly smaller contacts and lighter load per contact adhesion is more of an issue now and into the future.
2- Whilst the paper is scientifically very sound and interesting, its practical/application outlook is lacking somewhat. This is a pity, and I would like to encourage the author to provide some examples (even in passing comments) to make the readers more aware of the use of such relatively simple measures, i.e. one is not always required to carry out in-depth and time-consuming numerical analysis. This has been one of the main message of measures such as plasticity index.
3- It would also be interesting to state all the assumptions made in the model, such as isothermal, dry contact, etc.
The above 2 suggestions are not really prescriptive, but suggestions.
4- There is a fair amount of mathematical discourse which to an uninitiated young researcher can be foreboding and difficult to follow, particularly that no nomenclature is provided. Please include a full nomenclature of all the mathematical symbols used in 2 lists: one R Oman Symbols and the other Greek symbols. The nomenclature should appear before the References.
I look forward to receiving a revised version. Please highlight all the changes/additions made.
Comments on the Quality of English Language
English grammar is generally fine.
Reviewer 2 Report
Comments and Suggestions for Authors
It is an interesting study on the assessment of asperity deformation state with the consideration of size-dependent hardness through a simple combination of the Persson elasto-plastic solution and the SGP hardness model. The obtained results conflict with the classical contact mechanics without the size effect. However, the study proposed by Song et al (JMPS 106 (2017) 1–14) provided a conclusion that can support the obtained results, i.e. with size dependent plasticity giving a value of contact force-to-area ratio that approaches the elastic value as the roughness increases. The study can provide useful guidance for the multiscale modeling of rough surface contact if smaller asperities are involved. Some suggestions are listed as follows:
1. Is it possible to provide a criterion for judging the possible elastic or plastic deformation of asperities based on the proposed plasticity index Eq. 13? Current common criterions for the transition from elastic to plastic deformation of an asperity are based on the macroscopic hardness without the consideration of size-dependent hardness.
2. The total contact area shown in Figs. 1-4 includes the elastic and plastic area. Suggest to give the method to calculate the total contact in Section 3 of the manuscript.
3. Some typo of the manuscript: Page 5 the fourth line “that”; Page 7 the second line “the ratio of”.
Round 2
Reviewer 1 Report
Comments and Suggestions for Authors
The author has addressed all my suggestions. I recommend the paper for publication.
Comments on the Quality of English LanguageThe English grammar is fine.
Reviewer 2 Report
Comments and Suggestions for Authors
The manuscript can be accepted in the present form.